# Development of Meso- and Macro-Pore Carbonization Technology from Biochar in Treating the Stumps of Representative Trees in Taiwan

**Shih-Chi Lee [1], Yutaka Kitamura [2], Chuan-Chi Chien [1,*], Chun-Shen Cheng [1], Jen-Hao Cheng [1], Shu-Hsien Tsai [1] and Chin-Cheng Hsieh [3]**

1  Central Region Campus, Industrial Technology Research Institute, Nantou 310401, Taiwan
2  Faculty of Life and Environmental Sciences, University of Tsukuba, Ibaraki 305-8577, Japan
3  Department of Biomechatronics Engineering, National Ping-Tung University of Science and Technology, Pingtung 912301, Taiwan
*  Correspondence: varian@itri.org.tw; Tel.: +886-7-3311658-809; Fax: +886-7-3310560

**Abstract:** This study uses the tree stumps of the three representative trees in Taiwan (*Leucaena leucocephala, Syzygium samarangense,* and *Ziziphus jujuba*) as the material source and recyclable oyster shell powder as an activator. A carbonization process for upgrading and recycling the tree stumps was developed with our homemade, digital-controlled, energy-saving carbonization system. First, the tree stumps are carbonized at a medium temperature of 500 °C and then heated to 900 °C for high-temperature carbonization, followed by the activation procedure as required. With our method, we can produce biochar with a high proportion of fixed carbon and a high proportion of meso- and macropores while maximizing the yield of wood vinegar. The specific surface area of the meso- and macropores can reach up to 70 m$^2$/g or more. The effect of different activation materials on the pore characteristics and specific surface area of biochar was carefully examined. It was found that both KOH and oyster shell powder is the ideal activator for producing biochar with a high proportion of meso- and macropores. The FTIR spectrum, CEC, and contents of the ordinary elements and heavy metals of the biochar were also reported. It is clear from the FTIR data that the absorption peaks of the overall spectrum of the three types of biochar after carbonization at high temperature are cleaner than those of biochar carbonized at low temperature. This research can promote the recycling of agricultural residues, enhance soil carbon sequestration, preserve fertilizers, and suppress diseases and pests, moving towards approaching the goal of net-zero carbon emissions.

**Keywords:** biochar; carbonization process; net-zero carbon emissions

## 1. Introduction

The "4p1000" Initiative of the Paris Agreement, proposed in 2015, emphasized the important role of land in global carbon reduction and carbon sequestration. Moreover, in line with the global carbon reduction, "recycling agricultural resources" is listed as an important policy [1].

In recent years, the development and utilization of materials of biological sources (biomass materials) have become more and more popular. Generally speaking, biomass refers to the collective name of various organisms formed by photosynthesis of carbon dioxide and water. Biomass in a broad sense includes plants, microorganisms, and animals that feed on them and their wastes [2–4]. In a narrow sense, biomass refers to crop straws, trees, livestock manure, and other wastes produced in agricultural production activities [5–9]. There are many kinds of biomass materials, each of which may have diverse components. Among various biomass materials, agricultural and forestry wastes are particularly suitable for preparing biochar.

Biochar is a carbonaceous, porous material produced from the pyrolysis of agricultural residues and solid wastes, and it has been widely used as a soil amendment. Recent research and development of biochar has focused on climate investigations, pollutant immobilization, soil improvement strategies, nutrient recovery, engineering material production, and sewage treatment [10–12]. Most studies report the positive effects of the nutritional value of biochar, which helps to improve plant growth and fertilizer use efficiency [13,14]. The renewability, low cost, high porosity, high surface area, and customizable surface chemistry of biochar have provided excellent examples of its great promise in a variety of engineering applications [15]. Biochar material is a solid product obtained after biomass undergoes a series of pyrolysis reactions in an anaerobic or anoxic atmosphere, followed by being treated accordingly with various procedures [16]. Biochar prepared from carbonization of biomass usually shows loose morphology with a large specific surface area. Compared with traditional carbon materials, the surface of biochar possesses a large number of oxygen-containing functional groups [17], which provides potential for development of carbon-based composites by combining with other substances. Therefore, biochar is increasingly favored by researchers [18]. Biochar materials have a series of superior properties and have a relatively wide range of applications. They have been used as solid fuels, soil amendments, adsorbents, and energy storage materials, among other uses. Biochar has a porous structure and can be mixed with soil to increase its porosity, which in turn increases the amount of air and water in the soil. Different from other organic substances in the soil, biochar contains more trace elements and has good chemical stability. It will not be effloresced or decomposed for a long time, so the fertility of the soil can be improved continuously [19]. Uzoma et al. applied biomass carbon as a fertilizer to corn fields and found that the yield of corn was significantly increased [20]. Mixing biomass carbon with traditional inorganic or organic fertilizers in a certain proportion can delay the release of nutrients in the fertilizer, which reduces the loss of nutrients and significantly improves the utilization rate of the fertilizer.

According to a recent result of Xiaomin Zhu et al. [21], the effects of biochar on microbial activity are rather diverse, and seven possible mechanisms have been proposed, as shown in Figure 1: (1) Biochar, with its porous structure, can act as a shelter for microorganisms; (2) free radicals and VOCs on biochar are toxic to certain soil microorganisms, inhibit soil-borne pathogens, and promote plant growth; (3) biochar can improve soil properties (such as physical and chemical conditions) and change the growth pattern of soil microorganisms; (4) biochar-induced changes in enzyme activities can affect soil element cycling related to microorganisms; (5) biochar can absorb and enhance the hydrolysis of signaling molecules, thereby interrupting microbial communication and changing microbial communities' structure; (6) biochar can enhance adsorption (through biochar surface functional groups) and degradation of soil pollutants (promoted by electron transfer between biochar, microorganisms, and pollutants), which can reduce the toxicity of pollutants to soil microorganisms; and (7) by absorbing nutrient cations through functional groups, biochar can improve soil CEC and maintain nutrients to promote microbial growth [21]. They argue that the interaction between biochar and soil microbes can alter microbial communities and their metabolic pathways, thereby altering soil processes, and that there are also interactions between different environmental influences. In summary, biological fertilizers and formulations using biochar carriers will be more concentrated and beneficial to the effective change of the related microbial communities.

This study aims to systematically evaluate the use of biochar as a potential carrier material for agrochemical and microbial delivery. The key parameters of biochar are essential to assess the potential of any material for transport purposes. Parameters such as physicochemical properties of biochar, mechanisms of adsorption and release of agrochemicals and microorganisms from biochar, comparative evaluation of biochar compared to other carrier materials, etc., can then provide insights into the development of new types of biochar in agricultural economics and environmental benefits of crop cultivation. At the same time, in order to use the biochar as a carrier of beneficial bacteria, the biochar was

properly modified and adjusted, and then its characteristics were analyzed to investigate the specifications for the microbial beneficial bacteria to be encapsulated into the porous biochar. In order to develop new-type biological fertilizers and agricultural material, after prediagnosing the future intelligent agricultural system for pest and disease prevention and control, we will provide friendly and organic biochar-based farming fertilizers and biological agents for pest and disease control.

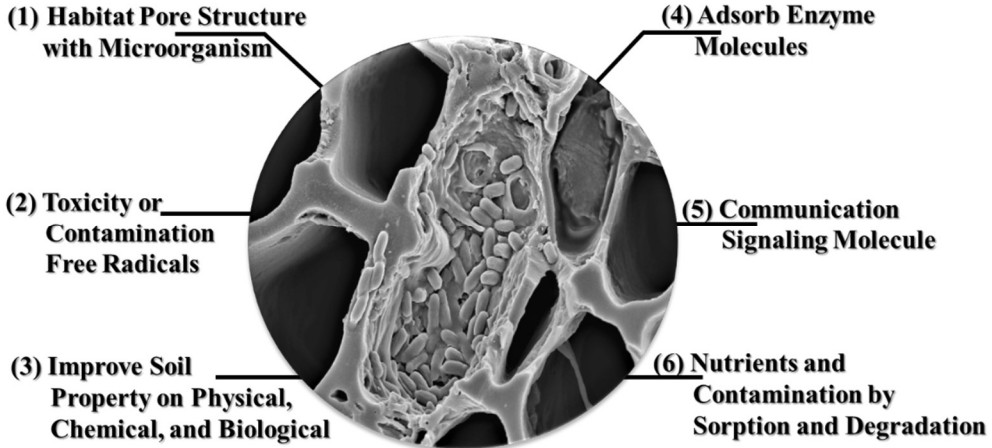

**Figure 1.** Functional analysis of biochar in soil.

However, as ZhongxinTan et al. [22] pointed out, there is a disconnect between biochar preparation and returning biochar to soil. To fully realize the potential of biochar in agriculture practices, that biochar researches must systematically consider from biochar preparation to its function in the soil.

In this study, we used tree stumps of the three representative trees in Taiwan, re-vealing a novel process for producing biochar with a high proportion of meso- and macropores and a high proportion of fixed carbon while maximizing the yield of wood vinegar and biochar (60%). The recycled oyster shell powder is found to be an ideal "green" activator for producing biochar with a high proportion of meso- and macropores. Biochar with a high proportion of meso- and macropores is an ideal inoculant for microbes. The biochar produced in this work can be further processed into various functional materials for agricultural applications.

## 2. Materials and Methods

### 2.1. Stumps of Three Representative Trees in Pingtung County, Taiwan: Leucaena leucocephala, Syzygium samarangense, and Ziziphus jujuba

Local name: White Popinac (*Leucaena leucocephala*), Local name: Wax apple (*Syzygium samarangense*), Local name: Dates (*Ziziphus jujube*). Activation material: oyster shell powder (from the oyster shell produced by Fangyuan Oyster Shell Factory); KOH (Potassium hydroxide) granular, purity $\geq$85%, molecular weight 56.11, CAS number: 1310-58-3, obtained from SIGMA, Germany, product number: 221473; $H_3PO_4$ (Phosphoric acid) liquid, purity $\geq$85%, molecular weight 98.00, CAS number: 7664-38-2, obtained from SIGMA, Germany, product number: 345245; Hydrochloric acid liquid, purity 30–50%, pH < 1 (20 °C), molecular weight 36.46, CAS number: 7647-01-0, obtained From SIGMA, Germany, Cat. No. H1758; Acetic acid liquid, purity $\geq$ 99%, pH 2.5 (20 °C, 50 g/L), MW 60.05, CAS No: 64-19-7, taken from SIGMA, Germany, Product No. A6283.

### 2.2. Carbonization Equipment: Energy-Saving Digital Carbonization System

The carbonization furnace used in this study is shown in Figure 2. This carbonization furnace was designed and manufactured by the Industrial Technology Research Institute. The main specifications of the equipment are as follows: for batch feeding, the carbonization

chamber volume is about 0.18 m³; feeding from the top and discharging from the lower side is adopted. According to the feeding size, moisture content, and specific gravity, each batch can fill 100~300 kg of biomass raw material. The equipment also has a wood vinegar collector, a hot water recovery system, and exhaust and filtration system. This unit can produce about 300–500 L of hot water at 45–60 °C per hour. The carbonization furnace does not use a blower for air intake, and no additional heat source is required except for ignition; the energy consumption per batch operation is only about 10–20 kWh. This equipment has the advantages of being energy-saving, having a high carbon conversion rate (25%), and producing a high wood vinegar yield (30%).

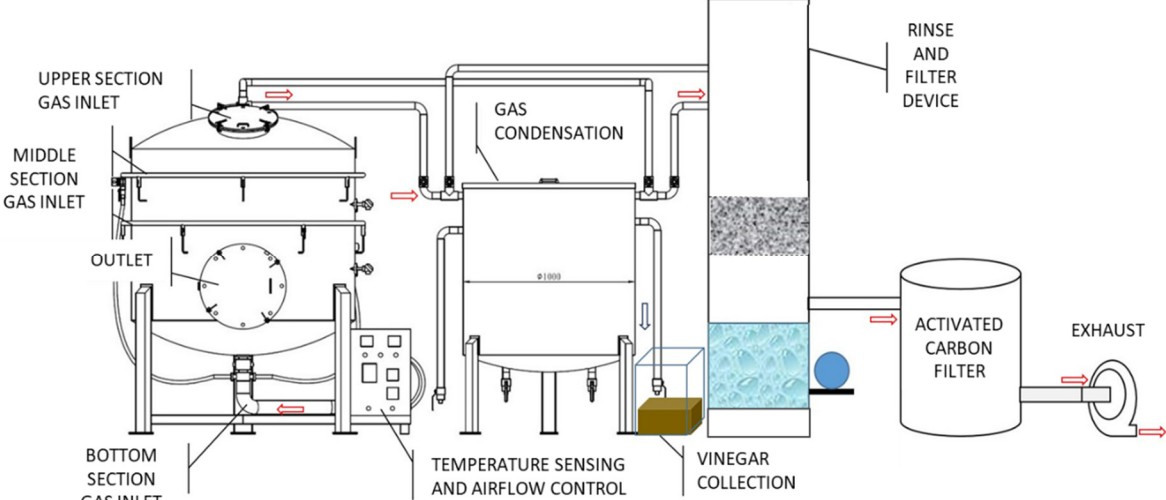

**Figure 2.** The energy-saving digital carbonization system.

### 2.3. Biochar Preparation

In this study, a two-step method was used for carbonization. All fruit tree stumps were first carbonized at a medium temperature of 500 °C for one hour and then carbonized at a high temperature of 900 °C for half an hour in order to improve the biochar yield. Natural cooling was adopted, and it took about 8 h to cool from 900 °C to room temperature [23].

### 2.4. Biochar Activation

Oyster shell powder (obtained from the oyster shell produced by Fangyuan Oyster Shell Factory) was used as a carbon dioxide activation modifier. The weight ratio of oyster shell powder to biochar was 1:1, and the activation was carried out at 900 °C for 60 min. KOH and $H_3PO_4$ were used as activation modifiers by mixing 20 g of the two activators and 40 mL of deionized water to form an activation solution. The activation solution and biochar were mixed at a weight ratio of 1:1, drained, and activated at 900 °C for 60 min.

### 2.5. Adjustment of the Acidity of Biochar

The biochar pH was adjusted with 1.0 M acetic acid. The adjustment method was to weigh about 45.5 g of biochar, add 20 times the weight of deionized water (910 mL), and evenly mix, then add 1.0 M acetic acid for acid–base neutralization until the pH value of the aqueous solution equaled 7.

### 2.6. Cation Exchange Capacity (CEC)

Biochar cation exchange was measured using the method of Matthew et al. [24]. The method is mainly based on the AOAC 973.09 method with minor adjustments.

### 2.7. Fourier Transform Infrared Spectroscopy (ATR-FTIR)

Biochar samples were detected by ATR-FTIR using a Jasco FT-IR 4100 spectrometer. After the biocarbon was ground in an agate mortar, it was directly placed on the measuring table for FTIR measurement in a reflective manner [25].

### 2.8. Mercury Intrusion Porosimetry

We used mercury porosimetry to measure pore volume and pore volume distribution for all biochar samples. Mercury porosimetry is suitable for measuring samples with a pore size range of 2.5 nm–400 μm (https://www.astm.org/d4404-18.html, accessed on 20 March 2018). We refer to the determination using ASTM D4404-18.

## 3. Results and Discussion

### 3.1. Analysis and Discussion of Pore Characteristics and Specific Surface Area at Different Carbonization Temperatures

Table 1 shows the measurement results of the pH value and related pore properties of the biochar produced by carbonization of three kinds of fruit tree stumps (*Leucaena leucocephala*, *Ziziphus jujuba,* and *Syzygium samarangense*) at different temperatures. All samples were first carbonized at 500 °C for 60 min and then heated to 900 °C for 60 min of high-temperature carbonization. The pH value of biochar samples fell between 9.64 and 10.03, and the pH value of *Syzygium samarangense* charcoal was the highest, which may be due to the high proportion of ash contained in *Syzygium samarangense* charcoal (see Table 2). According to the literature, when the carbonization temperature is lower than 900 °C, the main components of wood ash are $CaCO_3$ and $K_2Ca$ $(CO_3)_2$, and trace elements such as phosphorus, iron, manganese, zinc, and copper exist [26]. These elements are essential fertilizers for crops, and biochar with high ash content (Such as *Syzygium samarangense* charcoal) can be applied to the soil to help reduce the use of chemical fertilizers. It can be seen from Table 1 that high-temperature carbonization helps to increase the total surface area of biochar, which is consistent with the literature results [27]. Among the three tree species, the increase ratio of surface area was the largest in *Ziziphus jujuba*, followed by *Leucaena leucocephala*, and the least in *Syzygium samarangense*. It is worth noting that except for R9 (*Ziziphus jujuba* charcoal), the proportion of mesopores and macropores in the rest of the biochar samples is greater than 50% (according to the definition of IUPAC, pores with a pore size less than 2 nm are micropores, those with a pore size of 2~50 nm are mesopores, and those larger than 50 nanometers are macropores. In addition, since the effective measurable pore size range covered by the mercury intrusion method is about 2.5 nm–400 μm, the meso- and macropores referred to in this paper generally correspond to a pore diameter between 2 and 400 nanometers.) The ratio of surface area to volume of macropores in the biochar after high temperature treatment was found to decrease. A close check of the data in Table 1 reveals that, compared with the data at 500 °C, after the biochar was carbonized at a high temperature of 900 °C, the surface area and volume of macropores did not change significantly. In contrast, the surface area and volume of micropores increased significantly after high-temperature carbonization. Therefore, the ratio of surface area to volume of macropores in the biochar after high-temperature treatment is due to the increase of micropores caused by high-temperature carbonization, which also tells us that high-temperature carbonization does not significantly affect the meso- to macropore structure of biochar. These experimental results are consistent with those reported in the literature [28]. Interestingly, the effects of high-temperature carbonization on the microporous structure of different tree species are not the same. In terms of the decrease in the ratio of surface area to volume of meso- and macropores (the proportion of micropores increased), *Ziziphus jujuba* biochar is the most significant. The ratio decreased from 66.00% at 500 °C to 26.70% at 900 °C; the change of *Syzygium samarangense* biochar was second—the proportion of meso- to macro-pore surface area decreased from 98.52% at 500 °C to 58.46% at 900 °C, the proportion of meso- to macro-pore volume decreased from 72.18% at 500 °C to 57.36% at 900 °C; *Leucaena leucocephala* biochar showed the smallest change, in which

the ratio of macro-pore surface area decreased from 97.17% at 500 °C to 54.72% at 900 °C, and the volume ratio of meso- and macro- pores dropped from 83.10% at 500 °C to 81.62% at 900 °C; *Leucaena leucocephala* biochar showed the smallest change, in which the ratio of macropore surface area decreased from 97.17% at 500 °C to 54.72% at 900 °C; and the volume ratio of medium and large pores dropped from 83.10% at 500 °C to 81.62% at 900 °C.

**Table 1.** The pore characteristic parameters of the biochar produced by the carbonization process of three kinds of tree stumps (*Leucaena leucocephala*, *Ziziphus jujuba*, *Syzygium samarangense*) at 500 °C and 900 °C, respectively.

| Sample Source | Carbonization Temp. (°C) | Carbonization Period (min.) | pH | Specific Surface Area (SSA) (m²/g) | Microporous SSA (m²/g) | Meso & Macroporous SSA (m²/g) | Total Pore Volume (mL/g) | Microporous Volume (mL/g) | Meso & Macroporous Volumn (mL/g) | Meso & Macroporous SSA % | Meso & Macroporous Volumn % |
|---|---|---|---|---|---|---|---|---|---|---|---|
| *Leucaena leucocephala* | 500 | 60 | 9.94 | 35.79 | 0.82 | 34.97 | 0.0775 | 0.0131 | 0.0644 | 97.71 | 83.10 |
| | 900 | 30 | 9.98 | 82.46 | 37.34 | 45.12 | 0.0926 | 0.017 | 0.0756 | 54.72 | 81.64 |
| *Ziziphus jujuba* | 500 | 60 | 9.64 | 68.65 | 15.43 | 53.22 | 0.0947 | 0.0322 | 0.0625 | 77.52 | 66.00 |
| | 900 | 30 | 9.69 | 325.4 | 270.58 | 54.82 | 0.2232 | 0.1636 | 0.0596 | 16.85 | 26.70 |
| *Syzygium samarangense* | 500 | 60 | 10 | 67.57 | 1.00 | 66.57 | 0.1089 | 0.0303 | 0.0786 | 98.52 | 72.18 |
| | 900 | 30 | 10.03 | 102.79 | 42.70 | 60.09 | 0.1243 | 0.053 | 0.0713 | 58.46 | 57.36 |

**Table 2.** Fixed carbon content (%) in the biochar after physical activation at 900 °C.

| Sample Source | Volatile Components (%) | Fixed Carbon (%) | Ash (%) | pH |
|---|---|---|---|---|
| *Leucaena leucocephala* | 7.4 | 86.8 | 5.8 | 9.98 |
| *Ziziphus jujuba* | 5.5 | 90.1 | 4.4 | 9.69 |
| *Syzygium samarangense* | 4.6 | 86.0 | 9.4 | 10.03 |

Table 2 shows the ratio of volatility, fixed carbon (nonvolatile) and ash content of biochar samples of three kinds of fruit tree stumps treated at the high temperature of 900 °C. The data show that the biochar treated at 900 °C high temperature has a fixed carbon ratio of more than 86% and a volatile matter ratio of less than 8%, indicating that the temperature of 900 °C is enough to almost completely carbonize the three fruit tree stumps. Among the three kinds of biochar, the volatile components of the *Syzygium samarangense* carbon were the least, followed by the *Ziziphus jujuba* carbon, and the *Leucaena leucocephala* was the highest. In terms of ash, *Syzygium samarangense* has the highest carbon content. Based on the above experimental results, *Syzygium samarangense* biochar has the lowest volatile matter content and the highest ash content, which also shows that when *Syzygium samarangense* carbon is used as a soil conditioner, its safety is higher than the other two types of biochar. In addition, the experimental results listed above also show that the pore characteristics of biochar will be significantly different due to different biomass sources. It is because of this factor that biochar has so far been mostly used only as an inexpensive soil amendment [29].

### 3.2. The Effect of Different Activation Methods on the Pore Characteristics and Specific Surface Area of Biochar Samples

Due to the larger specific surface area of biochar, it can absorb more water or nutrients, and it has a better effect when used as a soil conditioner. In addition, from the above data, it is shown that in addition to *Ziziphus jujuba* biochar, the specific surface area of *Syzygium samarangense* biochar and *Syzygium samarangense* biochar after high temperature carbonization is only about 100 m²/g. In order to increase the specific surface area of biochar, we used three activators to activate the three types of biochar in this work at 900 °C. The three activators were: oyster shell powder, KOH, and $H_3PO_4$, and all biochar were activated at 900 °C for 60 min. Table 3 shows the detailed measurement data of the pore characteristics of the three types of biochar activated by oyster shell powder, KOH, or $H_3PO_4$, respectively. We first observed the effect of oyster shell powder activation on the

pore properties of biochar. Figure 3 shows the comparison of the BET specific surface area of the biochar samples of the three fruit tree stumps after low temperature carbonization, high temperature carbonization, and oyster shell activation. First, it can be seen from Figure 3 that the oyster shell powder significantly improved the specific surface area of the three biochar samples. Compared with high-temperature carbonization, the specific surface area of *Leucaena leucocephala* biochar increased by 2.81 times after activation by oyster shell powder, from 82.46 $m^2/g$ to 231.68 $m^2/g$; *Syzygium samarangense* biochar increased by 3.48 times from the original 102.8 $m^2/g$ to 357.91 $m^2/g$, and *Ziziphus jujuba* biochar increased from the original 325.4 $m^2/g$ to 330.96 $m^2/g$. Among the three kinds of biochar activated by oyster shells, the specific surface area of *Syzygium samarangense* charcoal is the largest, followed by *Ziziphus jujuba* charcoal, and *Leucaena leucocephala* charcaol is the smallest. Combining the three experimental results of the largest specific surface area, regarding the lowest volatile content and the highest ash content, *Syzygium samarangense* biochar once again shows that it is the most suitable for use as a soil conditioner. Next, we further observed the effect of oyster shell activation on the micropore and mesopore properties of biochar. Comparing Tables 1 and 3, it can be found that, except that the microporous surface area of *Ziziphus jujuba* carbon is slightly decreased after activation of oyster shell powder (from 270 $m^2/g$ to 258 $m^2/g$), the activation of oyster shell powder can be effective. At the same time, the micropores and the surface area of meso- and macro pores of biochar are improved. Further observation reveals that after the activation of oyster shell powder, the increase ratio of the micropore surface area is greater than that of the medium and large pores. Since micropores are pores with a pore size of less than 2 nm, increasing the surface area of micropores can improve the ability of biochar to adsorb small molecules (for example, water, inorganic salts, etc.); in contrast, increasing the surface area of medium and large pores of biochar, the ability of biochar to adsorb large molecules or substances, such as proteins or microorganisms, can be improved. The experimental data in Table 3 show that the specific surface area of mesoporous and macropores of *Leucaena leucocephala* carbon is 62.54 $m^2/g$ and that of *Ziziphus jujuba* carbon is 72.63 $m^2/g$. The data of medium-to-large pore surface area also echoes our previous conclusion that *Syzygium samarangense* carbon is most suitable as a soil amendment. Next, we observed the effect of KOH or $H_3PO_4$ activation on the pore properties of biochar samples. In order to compare the activation effects of the three activators on the specific surface area of BET, we cut out the data in Tables 1 and 3 and used the specific surface area of each biochar after carbonization at 900 °C as a normalization factor, which is plotted in the middle of Figure 4. First of all, it can be seen that different activators have different activation effects on the BET specific surface area of different species of biochar. For the three fruit tree stumps in this work, the activation effects of oyster shell powder and KOH are similar, and the increase in BET specific surface area is *Ziziphus jujuba* charcoal > *Leucaena leucocephala* charcoal > *Syzygium samarangense* charcoal, and the experimental results show that the activation effect of KOH is better if it is used in oyster shell powder; the surface area of *Syzygium samarangense* carbon activated by KOH is even close to 600 $m^2/g$. In contrast, the use of $H_3PO_4$ as an activator has a significant activation effect on *Leucaena leucocephala* charcoal and *Syzygium samarangense* charcoal. However, in the case of *Ziziphus jujuba* charcoal, its BET specific surface area decreases after activation. This indicates that $H_3PO_4$ may cause the collapse of micropores or closed pores in *Ziziphus jujuba* charcoal [30]. In terms of micropore surface area, the activation effect of KOH was better than that of oyster shell and $H_3PO_4$. In *Ziziphus jujuba* charcoal, we observed that the microporous surface area of *Ziziphus jujuba* carbon decreased significantly after $H_3PO_4$ treatment (the macroporous surface area also decreased, but the magnitude was not as large as that of the microporous surface area), which corresponds to the previous *Ziziphus jujuba* charcoal. After the carbon is activated by $H_3PO_4$, the BET specific surface area decreases. We can know that the decrease of the BET specific surface area of *Ziziphus jujuba* charcoal is mainly caused by the decrease of its micropore surface area. This paper focuses on the part discussed, so more in-depth research is needed to understand the mechanism of the reduction of micropore surface area. In terms

of the surface area of medium and large pores, the activation effects of the three activators are summarized in Figure 5 (the surface area of macropores in the unactivated biochar carbonized at 900 °C is used as the comparison basis). It can be clearly seen from Figure 5 that the use of oyster shell powder and KOH has a good activation effect on the medium and large pore surface areas of the three types of biochar, and the activation effect of KOH is better than that of oyster shell powder. Similar to the previous situation, using $H_3PO_4$ as an activator will also lead to a decrease in the specific surface area of medium and large pores of *Ziziphus jujuba* charcoal, and the activation effect of *Syzygium samarangense* charcoal is not good. Although in terms of micropore surface area the activation effect of KOH was better than that of oyster shell, and H3PO4 is better than that of oyster shell powder, because oyster shell powder is a natural carbon neutralization and recycling material, using oyster shell powder as an activator can reduce carbon emissions in the process and promote the environmental protection benefits of agricultural waste resource reuse. Finally, in terms of total pore volume, the effect of KOH and $H_3PO_4$ on the total pore volume of biochar is greater than that of oyster shell. Compared with KOH and $H_3PO_4$, the activation effect of oyster shell on total pore volume is poor. Compared with KOH and $H_3PO_4$, the effective active ingredient in the shell can produce less gas at high temperature, and the oyster shell powder is less likely to enter the small pores of the unactivated prebiochar because of its larger size [31,32].

**Table 3.** Measurement results of pore characteristics of each type of biochar after activation and modification (*n* = 3). The data of only high temperature carbonization but not activated are also attached to the reference for discussion.

| Sample Source | Carbonization Temp. (°C) | Carbonization Period (min.) | Specific Surface Area (SSA) (m²/g) | Microporous SSA (m²/g) | Meso and Macroporous SSA (m²/g) | Total Pore Volume (mL/g) | Microporous Volume (mL/g) | Meso and Macroporous Volumn (mL/g) | Meso and Macroporous SSA % | Meso and Macroporous Volumn % | Activator |
|---|---|---|---|---|---|---|---|---|---|---|---|
| W9-*L. leucocephala* | 900 | 30 | 82.46 | 37.34 | 45.12 | 0.0926 | 0.017 | 0.0756 | 54.72 | 81.64 | N/A |
| R9-*Z. jujuba* | 900 | 30 | 325.4 | 270.58 | 54.82 | 0.2232 | 0.1636 | 0.0596 | 16.85 | 26.70 | N/A |
| S9-*S. samarangense* | 900 | 30 | 102.79 | 42.70 | 60.09 | 0.1243 | 0.053 | 0.0713 | 58.46 | 57.36 | N/A |
| W9P-*L. leucocephala* | 900 | 60 | 231.683 | 169.150 | 62.540 | 0.192 | 0.108 | 0.084 | 26.99 | 43.71 | oyster shell powder |
| R9P-*Z. jujuba* | 900 | 60 | 330.960 | 258.330 | 72.630 | 0.248 | 0.219 | 0.029 | 21.95 | 11.52 | oyster shell powder |
| S9P-*S. samarangense* | 900 | 60 | 357.907 | 278.480 | 79.430 | 0.278 | 0.171 | 0.107 | 22.19 | 38.46 | oyster shell powder |
| W9K-*L. leucocephala* | 900 | 60 | 396.17 | 326.89 | 69.82 | 0.2782 | 0.1981 | 0.0801 | 17.62 | 28.79 | KOH |
| R9K-*Z. jujuba* | 900 | 60 | 433.94 | 345.88 | 88.06 | 0.3197 | 0.2172 | 0.1025 | 20.29 | 32.06 | KOH |
| S9K-*S. samarangense* | 900 | 60 | 575 | 498.11 | 76.89 | 0.3884 | 0.2968 | 0.0916 | 13.37 | 23.58 | KOH |
| W9H-*L. leucocephala* | 900 | 60 | 251.72 | 179.63 | 72.09 | 0.209 | 0.0898 | 0.1192 | 28.64 | 57.03 | $H_3PO_4$ |
| R9H-*Z. jujuba* | 900 | 60 | 144.89 | 103.81 | 41.08 | 0.1288 | 0.0544 | 0.0744 | 28.35 | 57.76 | $H_3PO_4$ |
| S9H-*S. samarangense* | 900 | 60 | 633.98 | 570.8 | 63.18 | 0.3796 | 0.2746 | 0.105 | 9.97 | 27.66 | $H_3PO_4$ |

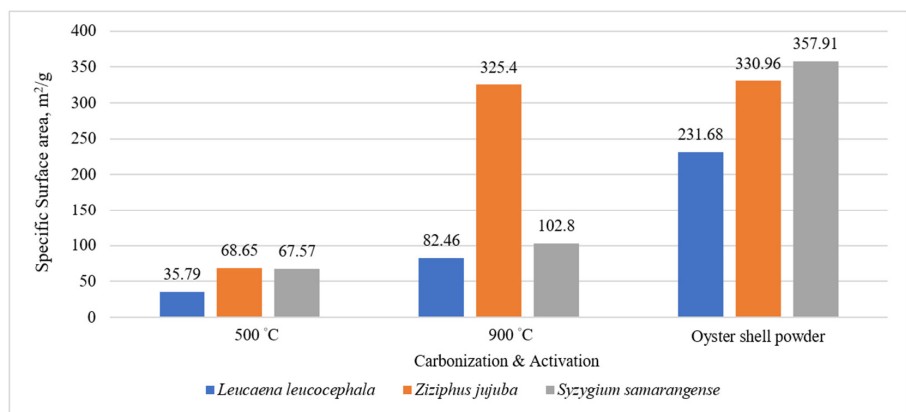

**Figure 3.** Effects of low temperature carbonization, high temperature carbonization and activation of oyster shell powder on the specific surface area of three fruit tree biochar samples.

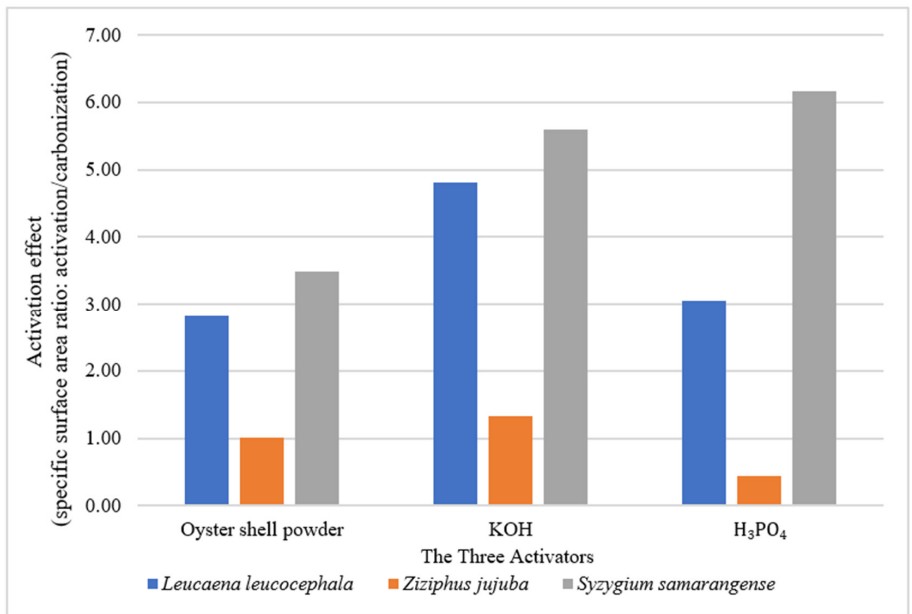

**Figure 4.** The activation effect of three activators on the specific surface area of three fruit tree stumps after high-temperature carbonization at 900 °C.

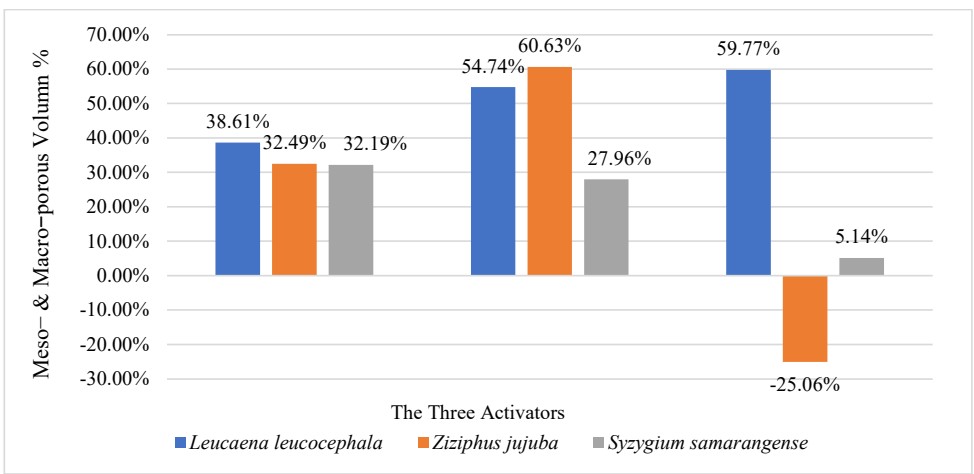

**Figure 5.** The surface area of the meso- and macropores of each type of biochar sample carbonized at 900 °C.

### 3.3. Analysis of the Properties of Biochar Fertilizers after Modification

The freshly prepared biochar has a high alkalinity, and it can be seen from Table 2 that the pH of the untreated biochar is about 10. If it is to be used as an encapsulating carrier for microorganisms for inoculation, the pH value of the biochar must be adjusted to neutral or weakly acidic to improve the survival rate and bacterial volume of the encapsulated microorganisms. In addition, due to the high solubility of metal ions in acid, the acid-washing process can also help to remove the residual heavy metals in the biochar [33]. In this work, therefore, 1.0 M acetic acid was used as the pickling solution. After weighing the three kinds of biochar materials after physical activation, 20 times the weight of water was added to mix them evenly, and then 1.0 M acetic acid aqueous solution was gradually added until the pH value of the aqueous solution reached neutrality (pH = 7.05). The experimental results showed (Figure 6) that to achieve neutrality, the amount of acetic acid used in the *Syzygium samarangense* biochar was the most, followed by the *Leucaena leucocephala*, and the *Ziziphus jujuba* the least. This means that the alkaline substance content in *Syzygium samarangense* charcoal is the highest. The results of this experiment are consistent with the results of ash content and pH value of biochar in Table 2. Table 4 shows the cation exchange capacity data of the three types of biochar [34]. The size of cation exchange capacity is *Ziziphus jujuba* charcoal > *Syzygium samarangense* charcoal > *Leucaena leucocephala* charcoal. The higher the cation exchange capacity, the more cations that the biochar can absorb, so the effect of using it as a kind of fertilizer is better. To explain the trend of the cation exchange capacity of the three types of biochar, we first observed that since the BET specific surface area of the *Leucaena leucocephala* carbon activated by the oyster shell was the smallest (Table 3), the smaller the specific surface area, the more effective the area that can be used to adsorb cations, in agreement with the fact that the Lactobacillus charcoal has the smallest cation exchange capacity. However, we noticed that the BET specific surface area of lotus charcoal activated by oyster shell powder was about 8% larger than that of *Ziziphus jujuba* charcoal (358 $m^2/g$ for *Syzygium samarangense* charcoal and 331 $m^2/g$ for *Ziziphus jujuba* charcoal), but its cationic exchange capacity was 28% less than that of *Ziziphus jujuba* charcoal, which means that the difference in cation exchange capacity between the types of biochar cannot be precisely explained by the BET surface area size alone. In view of this, we calculated the cation exchange capacity per unit BET surface area of the three types of biochar (the rightmost column of Table 5). The calculation results show that: *Ziziphus jujuba* charcoal > *Syzygium samarangense* charcoal ≈ *Leucaena leucocephala* charcoal, indicating the cation adsorption capacity per pore surface area of *Ziziphus jujuba* charcoal is the strongest. We speculate that this may be due to the presence of more chemical functional groups on the surface of *Ziziphus jujuba* charcoal that can adsorb cations. Comparing the FTIR spectra of the three types of biochar in the lower half of Figure 7, we can find that, compared with the *Leucaena leucocephala* charcoal and the *Syzygium samarangense* charcoal, the *Ziziphus jujuba* charcoal has the most obvious absorption peak at 1700 $cm^{-1}$. In the discussion later in this article, we will see that the absorption peak at 1700 $cm^{-1}$ comes from the C=O stretching vibration from the carboxylic group, which indicates that the pore surface of *Ziziphus jujuba* charcoal has more carboxyl groups (COOH). Because COOH is an anion site, it can generate electrostatic interaction with cations, which explains why *Ziziphus jujuba* charcoal has the largest cation exchange capacity per unit surface area.

**Table 4.** Cation exchange capacity (CEC) of each type of biochar after activation with oyster shells.

| Sample Source | CEC (cmol(+)/kg BC) | (CEC/BET) |
|---|---|---|
| *Leucaena leucocephala* | 6.8 | 0.0293 |
| *Ziziphus jujuba* | 13.3 | 0.04 |
| *Syzygium samarangense* | 10.4 | 0.0291 |

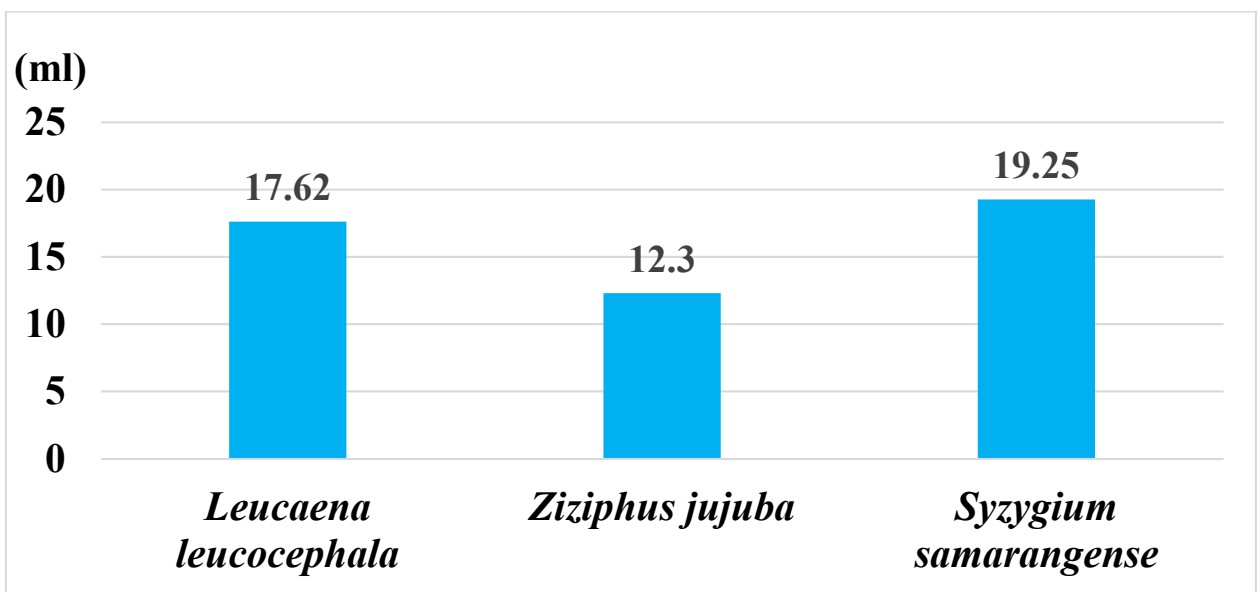

**Figure 6.** The pH values of 45.5 g biochar samples (1:20; carbon weight/deionized water) after acid washing and modification of 1.0 M acetic acid solution for three physical oyster shell activated biochar samples were measured after standing for one day, added, and adjusted to reach a stable pH = 7.05.

**Table 5.** The detection results of ordinary elements and heavy metals in the three biochar samples. The detection of heavy metals in biochar was carried out according to the European Biochar Certificate (EBC) specifications.

| Non-Heavy Metal | N % | P % | K% | Ca (ppm) | Mg (ppm) | Mn (ppm) | Fe (ppm) | B (ppm) | Na (ppm) |
|---|---|---|---|---|---|---|---|---|---|
| *Leucaena leucocephala* | 1.07 | 0.05 | 0.74 | 13,633.23 | 1838.62 | 22.27 | 187.26 | 53.02 | 677.77 |
| *Ziziphus jujuba* | 0.87 | 0.29 | 2.05 | 36,618.98 | 4692.50 | 72.57 | 188.10 | 31.81 | 1213.07 |
| *Syzygium samarangense* | 1.01 | 0.24 | 1.16 | 25,232.43 | 3471.38 | 25.43 | 145.21 | 46.83 | 1162.53 |
| **Heavy metal** | Cu (ppm) | Zn (ppm) | Cd (ppm) | Cr (ppm) | Ni (ppm) | Pb (ppm) | As (ppm) | Hg (ppm) | |
| *Leucaena leucocephala* | 6.40 | 12.01 | <0.009 | 32.83 | 20.72 | <0.027 | <0.005 | 0.30 | |
| *Ziziphus jujuba* | 23.57 | 32.05 | <0.009 | 37.40 | 26.85 | <0.027 | <0.005 | 0.52 | |
| *Syzygium samarangense* | 14.70 | 12.85 | <0.009 | 6.52 | 5.70 | <0.027 | <0.005 | 0.52 | |
| Basic (Max.) | 100.00 | 400.00 | 1.50 | 90.00 | 50.00 | 150.00 | 13.00 | 1.00 | |
| Premium (Max.) | 100.00 | 400.00 | 1.00 | 80.00 | 30.00 | 120.00 | 13.00 | 1.00 | |

Table 5 shows the contents of different metals in the three types of biochar. It can be seen that the top five metals with the highest contents in the biochar are calcium, potassium, magnesium, sodium, and iron. Among these five metals, the content of *Ziziphus jujuba* carbon is the highest, followed by *Syzygium samarangense* carbon, and Lactobacillus is the least, which is in line with the trend of the cation exchange capacity of the three types of biochar (the higher the cation exchange capacity of the biochar, the higher the number of cations it can adsorb). Calcium, potassium, magnesium, sodium, and iron are all common elements that are indispensable for plant growth. The experimental results also show that *Ziziphus jujuba* charcoal and *Syzygium samarangense* charcoal are more suitable for

use as quasi-fertilizers than *Leucaena leucocephala* charcoal. In addition, if biochar is to be used as a quasi-fertilizer, the heavy metals contained in it must meet the specifications. We have attached the upper limit of heavy metal content of European biochar EBC in Table 5. EBC regulates copper, zinc, cadmium, nickel, lead, arsenic, and mercury, which are seven categories of heavy metals, and biochar is classified into two categories: basic and high-quality, according to their content. The experimental results in Table 5 show that the heavy metal content of biochar made from three common waste branches in Taiwan meets the European EBC high-quality biochar standard.

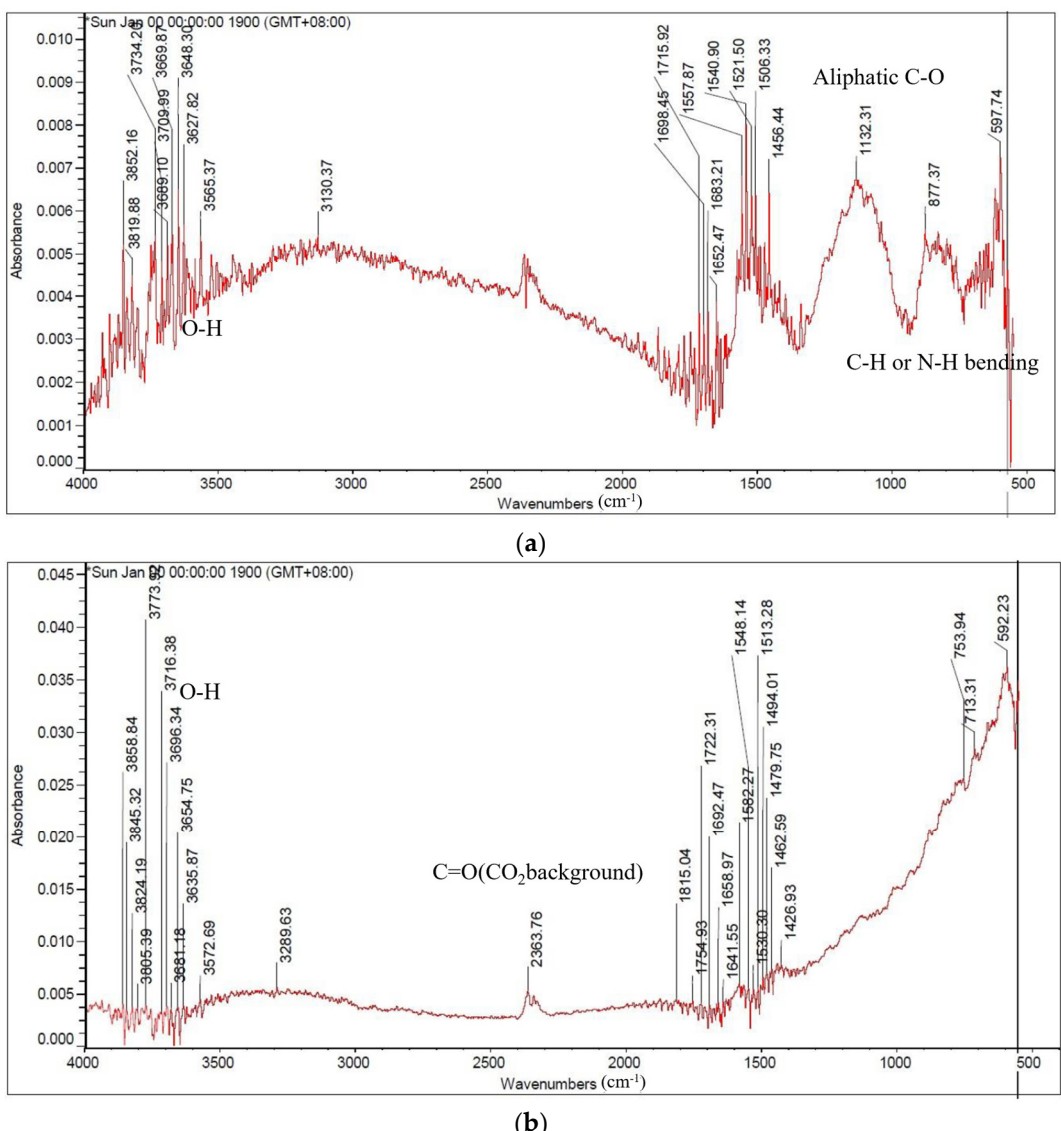

(**a**)

(**b**)

**Figure 7.** *Cont.*

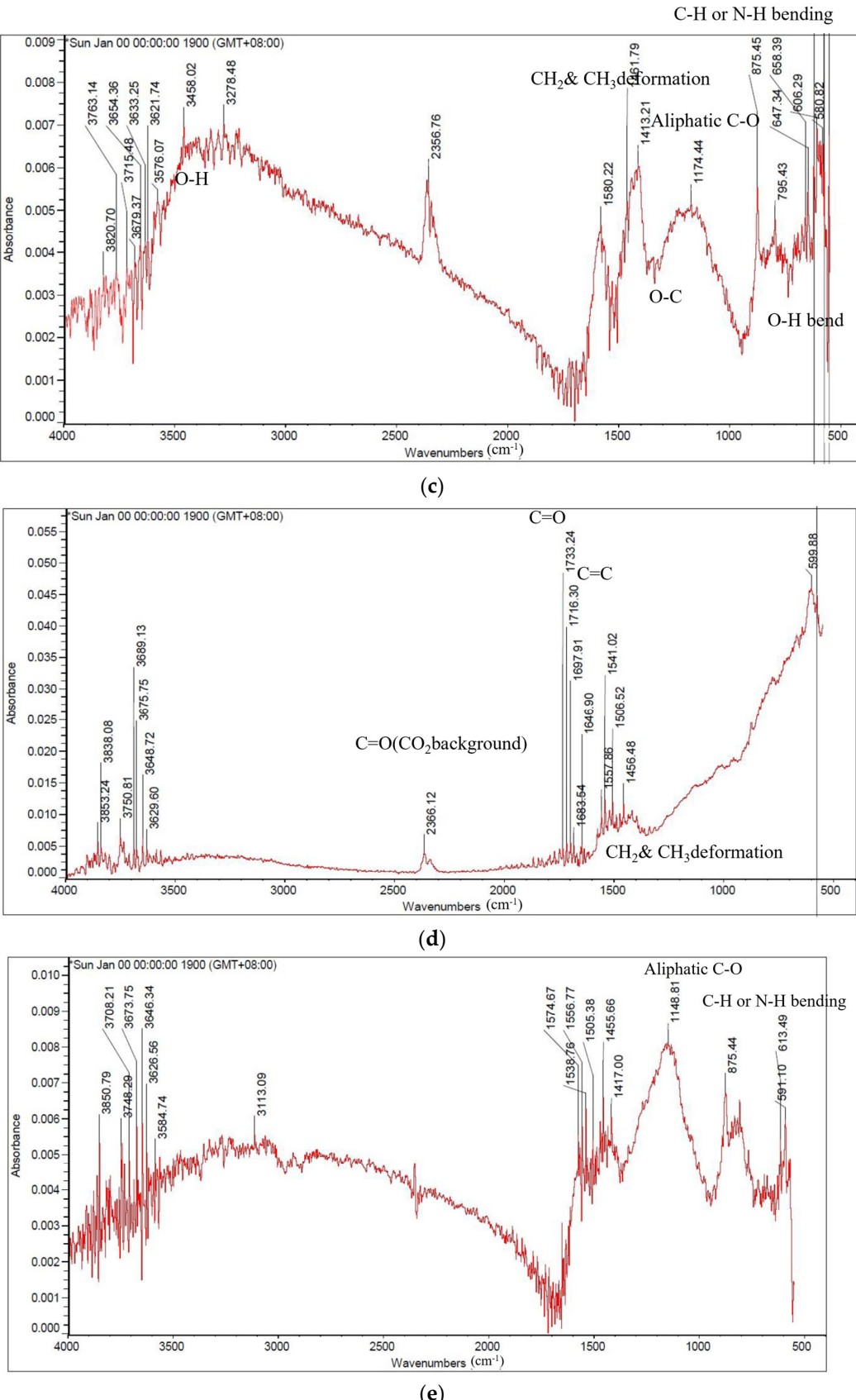

**Figure 7.** *Cont.*

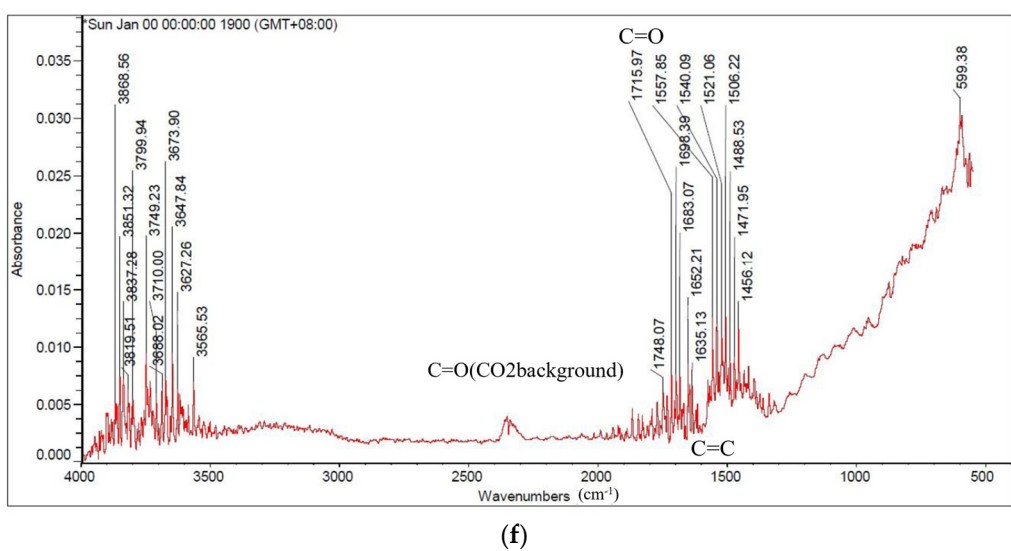

(**f**)

**Figure 7.** The IR vibrational wavenumbers of the functional groups of the *Leucaena leucocephala* (**a**,**b**), *Syzygium samarangense* (**c**,**d**), *Ziziphus jujube* (**e**,**f**) stump biochar produced by activation and modi-fication at 500 and 900 °C.

### 3.4. Investigate the Degree of Carbonization by Functional Group Analysis by FTIR

Figure 7 is the infrared spectrum data of different biochar materials after three different fruit tree stumps were carbonized at 500 °C and activated at a high temperature of 900 °C. First of all, it is clear from the FTIR data that the absorption peaks of the overall spectra of the three types of biochar after carbonization at high temperature are cleaner than those of biochar carbonized at low temperature, which is consistent with the literature results that high temperature can improve the degree of carbonization [35,36]. According to the literature, biochar mainly has the following infrared characteristic peaks: the absorption peak near 3400 cm$^{-1}$ comes from the stretching vibration (stretching) of the O–H bond in alcohols or phenolic compounds; the absorption peak near 2900 cm$^{-1}$ comes from the C–H stretching vibration of aliphatic hydrocarbons generated during carbonization; the absorption peaks around 2830–2670 cm$^{-1}$ correspond to the bending vibrations (bending) of aliphatic hydrocarbons generated during carbonization; the absorption peak around 1700 cm$^{-1}$ comes from C=O stretching vibration of carboxylic group; the absorption peaks around 1600 cm$^{-1}$ are from -C=C- stretching vibration on hemicellulose; the absorption peaks at 1120–1050 cm$^{-1}$ are from the C–O stretching vibration of cellulose and hemicellu-lose; [37]. As a side note, the phenomenon spectrally located at 2360 cm$^{-1}$ arises from the absorption peaks of the C–O asymmetric stretching of carbon dioxide molecules in the ambient background, which has nothing to do with the surface properties of biochar. Specifically, the biochar of the three fruit tree stumps is around 3400 cm$^{-1}$. The absorption peak of OH decreased significantly after carbonization at 900 °C high temperature, indicating that high temperature helps to convert lignin in tree stumps into fixed carbon. The OH absorption peak intensity of *Leucaena leucocephala* and *Ziziphus jujuba* biochar carbonized at 500 °C is significantly lower than that of *Syzygium samarangense* carbon carbonized at 500 °C; the IR peaks of the three kinds of fruit tree stump samples are still visible at 2900–2670 cm$^{-1}$ after carbonization at 500 °C for one hour, indicating that 500 °C carbonization for one hour is not enough to make the materials (lignin, fibers and hemicelluloses) completely carbonized; in all biochar samples carbonized at 500 °C, there were significant absorption peaks around 1700 cm$^{-1}$, because the absorption peaks in this region corresponded to the C=O stretching vibration of the carboxylic group. These experimental results show that at 500 °C carboniza-tion has begun to produce acid (wood vinegar). Since the higher carbonization temperature will lead to the decomposition of the acid solution, in order to improve the yield of wood vinegar during the pyrolysis process, it is not suitable to collect the vinegar solution at an excessively high carbonization temperature; all the absorption peaks in the range of

$1600 \text{ cm}^{-1}$ to $1050 \text{ cm}^{-1}$ come from cellulose or hemicellulose and its derivatives. As can be seen from Figure 6, after the three sources are carbonized at 500 °C, a very significant signal of cellulose or hemicellulose derivatives can still be observed. However, after carbonization at 900 °C, these signals were quite weak, and the same phenomenon also occurred in $895–880 \text{ cm}^{-1}$ from the stretching vibration signals from olefins and $880–720 \text{ cm}^{-1}$ from the signals from aromatic compounds, indicating that 900 °C is sufficient for the three fruit trees in this work. The stumps are completely carbonized. Even though aromatic hydrocarbons have better chemical stability than aliphatic hydrocarbons, the carbonization process at 900 °C for one hour is still enough to eliminate aromatic hydrocarbons generated during the cracking process. Since most aromatic hydrocarbons are biologically toxic, carbonization processes above 500 °C can help eliminate harmful aromatic hydrocarbons in biochar for the purpose of microbial encapsulation and soil improvement (Table 6).

**Table 6.** The functional groups of biochar from the stump representative trees in Taiwan that already was increased or decreased by activation and modification at 500 and 900 °C.

| The Stumps of Representative Trees in Taiwan | Carbonization Temperature (°C) | Absorption Peak (cm$^{-1}$) | | | | | | | |
|---|---|---|---|---|---|---|---|---|---|
| | | 3400 | 29.00 | 2830–2670 | 1700 | 1600 | 1480–1410 | 1120–1050 | 895–880 |
| | | Alcohols or Phenolic | Aliphatic Hydrocarbons | | Carboxylic Group | Hemi-Cellulose | Cellulose & Hemicellulose | | Olefins |
| | | O-H Stretching | C-H Stretching | C-H Bending | C=O Stretching | C=C-Stretching | C-H Deformation | C-O Stretching | C=C Stretching |
| *Leucaena leucocephala* | 500 | + | + | + | + | + | + | + | + |
| | 900 | - | - | - | - | - | - | - | - |
| *Syzygium sama-rangense* | 500 | + | + | + | + | + | + | + | + |
| | 900 | - | - | - | - | - | - | - | - |
| *Ziziphus jujuba* | 500 | + | + | + | + | + | + | + | + |
| | 900 | - | - | - | - | - | - | - | - |

+: With functional groups; -: Decreased functional groups.

## 4. Conclusions

In this work, a method for producing biochar with a high proportion of meso- and macropores and high proportion of fixed carbon was developed. A self-developed, digital, energy-saving carbonization system was employed in this work. The tree stumps of the three representative trees in Taiwan (*Leucaena leucocephala, Syzygium sa-marangense, and Ziziphus jujuba*) were used as the material source. A two-step carbonization process was performed to obtain high-yield wood vinegar (30%) and high-quality biochar. It was found that both KOH and oyster shell powder were good activators for increasing the specific surface area of the meso- and macropores of the biochar samples. For the three kinds of tree stump biochar activated by oyster shell powder, the specific surface area of meso- and macropores can be greater than 70 m$^2$/g. The results of heavy metal measurement show that the three types of biochar meet the European EBC high-quality biochar standard. The results of FTIR showed that the volatile carbon content of the biochar samples was significantly reduced after 900 °C carbonization. The biochar with high specific surface area of the meso- and macropores is a good inoculant material for microbes and can act as a slow-released fertilizer for various agricultural applications.

**Author Contributions:** Conceptualization, C.-C.C.; Data curation, C.-S.C. and J.-H.C.; Formal analysis, C.-S.C.; Funding acquisition, S.-C.L. and C.-C.C.; Investigation, S.-C.L., C.-C.C. and S.-H.T.; Project administration, S.-C.L. and C.-C.C.; Supervision, Y.K., C.-C.C. and C.-C.H. All authors have read and agreed to the published version of the manuscript.

**Funding:** Technology Development Program (TDP), Department of Industrial Technology, Ministry of Economic Affairs, R.O.C.

**Data Availability Statement:** The study did not report any data.

**Conflicts of Interest:** The authors declare no conflict of interest.

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
