# Peer review of "Development of Meso- and Macro-Pore Carbonization Technology from Biochar in Treating the Stumps of Representative Trees in Taiwan"

_sustainability, doi:10.3390/su142214792_

Round 1

Reviewer 1 Report

Manuscript ID: sustainability-1935282: Development of meso- and macro-pore carbonization technology in treating the stumps of representative trees in Taiwan

1.      abstract should be informative, quantitative and also remove the grammatical errors?

2.      Novelty of study is missing in the introduction section?

3.      In introduction citation should be improve with recent literatures?

4.      Characterization part of manuscript is very weak so improved through addition such as SEM, EDS, TGA and XRD etc?

5.      How to know Carbonization temperature (oC) of sample shown in table 1 and discuss about that one paragraph?

6.      Citation should be update with recent study? 3.2 The effect of different activation methods on the pore characteristics and specific surface area of biochar samples, compared with KOH and H3PO4, the effective active ingredient in the shell can produce less gas at high temperature and the oyster shell powder is less likely to enter the small pores of the unactivated pre-biochar because of its larger size.

7.      In section 3.2 The effect of different activation methods on the pore characteristics and specific surface area of biochar samples, Author showed these results (Table 3 show that the specific surface area of mesoporous and 267 macropores of Leucaena leucocephala carbon is 62.54 m2 /g; that of Ziziphus jujuba carbon is 268 72.63 m2 /g.) but they should compare with literatures also for scientifically?

8.      in section 3.3 Analysis of the properties of biochar fertilizers after modification, (“After weighing the three kinds of biochar materials after physical activation, 20 times the weight of water was added to mix them evenly, and then 1.0 M acetic acid aqueous solution was gradually added until the pH value of the aqueous solution reached neutrality (pH=7.05).”.) this parameter and also compare with recent literatures?

9.      Figure 7. The IR vibrational wavenumbers of the functional groups of the Leucaena leucocephala stump biochar produced by activation and modification at 500°C (W5-f) and at 900°C (W912), should be merged in single figure so that difference will, see?

10.  Figure 8. The IR vibrational wavenumbers of the functional groups of the Syzygium samarangense stump biochar produced by activation and modification at 500°C (S5-f) and at 900°C (S912), should be merged in single figure so that difference will, see?

11.  Figure 9. The IR vibrational wavenumbers of the functional groups of the Ziziphus jujuba stump 409 biochar produced by activation and modification at 500°C (I5-f) and at 900°C (I912), should be merged in single figure so that difference will, see?

12.  Citation of result discussion will be improved through recent studies in field?

13.  Conclusion of manuscript will be concise and with quantitative results?

Author Response

Manuscript ID: sustainability-1935282: Development of meso- and macro-pore carbonization technology in treating the stumps of representative trees in Taiwan

Many thanks to the reviewer for valuable suggestions. The relevant additions and revisions are as follows:

  1. abstract should be informative, quantitative and also remove the grammatical errors?

A: Thank you very much for your valuable suggestions, which have been revised. Please See page 1, line 13 to 30.

  1. Novelty of study is missing in the introduction section?
  2. In introduction citation should be improve with recent literatures?

A: Thank you very much for the valuable suggestions of the committee members, and the description of novelty has been added. The development of macro porous biochar as a carrier for probiotics is quite an innovative idea. See page 3, line 109 to 120.  Added to recent literatures [30][36][37].

[30] Zhongxin Tan, Carol S.K.Lin, Xiaoyan Ji, Thomas J. Rainey. (2017). Returning biochar to fields: A review. Applied Soil Ecology, 116, 1-11.

[36] Laghari, M.; Naidu, R.; Xiao, B.; Hu, Z.; Mirjat, M. S.; Hu, M.; Kandhro. M. N.; Chen. Z.; Gau. D.; Jogi.; Abudi. Z. N.; Fazal. S. (2016). Recent developments in biochar as an effective tool for agricultural soil management: a review. J. Sci. Food Agric. 2016, 96, 4840-4849.

[37] Semida, W. M.; Beheiry, H. R.; Sétamou, M.; Simpson, C. R.; Abd El-Mageed, T. A.; Rady, M. M.; Nelson, S. D. Biochar implications for sustainable agriculture and environment: A review. S. Afr. J. Bot. 2019, 127, 333-347.

  1. Characterization part of manuscript is very weak so improved through addition such as SEM, EDS, TGA and XRD etc?

A: Thank you very much for your valuable suggestions.

This paper is "Development of meso- and macro-pore carbonization technology in treating the stumps of representative trees in Taiwan".

Therefore, at the stage of this paper, the carbonization process and the properties of pores are analyzed to understand the distribution and proportion of marco-, meso- and micro pores after carbonization and activation of these three representative tree branches in Taiwan.

Development of macro porous biochar loaded probiotics as the next stage. Therefore, SEM, EDS, TGA and XRD measurements are not required at this stage.

  1. How to know Carbonization temperature (oC) of sample shown in table 1 and discuss about that one paragraph?

A: Thank you very much for your valuable suggestions.

P.5 to 6, line 180 to 224. “3.1 Analysis and discussion of pore characteristics and specific surface area at different carbonization temperatures.” The entire article describes the carbonization temperature and porosity characteristics of the three materials.

  1. Citation should be update with recent study? 3.2 The effect of different activation methods on the pore characteristics and specific surface area of biochar samples, compared with KOH and H3PO4, the effective active ingredient in the shell can produce less gas at high temperature and the oyster shell powder is less likely to enter the small pores of the unactivated pre-biochar because of its larger size.
  2. In section 3.2 The effect of different activation methods on the pore characteristics and specific surface area of biochar samples, Author showed these results (Table 3 show that the specific surface area of mesoporous and 267 macropores of Leucaena leucocephala carbon is 62.54 m2 /g; that of Ziziphus jujuba carbon is 268 72.63 m2 /g.) but they should compare with literatures also for scientifically?

A: Thank you very much for your valuable suggestions.

Two new papers have been added for reference.

[31]Abdul Hafeez, Taowen Pan, Jihui Tian, & Kunzheng Cai. (2022). Modified Biochars and Their Effects on Soil Quality: A Review. Environments 9(5):60, DOI:10.3390/environments9050060.

[32]Wang, Y.; Zhong, B.; Shafi, M.; Ma, J.; Guo, J.; Wu, J.; Ye, Z.; Liu, D.; Jin, H. Effects of biochar on growth, and heavy metals accumulation of moso bamboo (Phyllostachy pubescens), soil physical properties, and heavy metals solubility in soil. Chemo-sphere 219, 510–516.

  1. in section 3.3 Analysis of the properties of biochar fertilizers after modification, (“After weighing the three kinds of biochar materials after physical activation, 20 times the weight of water was added to mix them evenly, and then 1.0 M acetic acid aqueous solution was gradually added until the pH value of the aqueous solution reached neutrality (pH=7.05).”.) this parameter and also compare with recent literatures?

A: Thank you very much for your valuable suggestions.

Biochar for soil improvement or combination of probiotics, biochar must have a suitable pH. For biochar-coated probiotics, the pH should be neutral, so the pH adjustment of relevant biochar can refer to this book [33].

[33]Balwant Singh, Dolk MM, Qinhua Shen, Marta Camps Arbestain. (2017). Chapter 3. Biochar pH, electrical conductivity and liming potential. In book: Biochar: A Guide to Analytical Methods (pp.23-38).

  1. Figure 7. The IR vibrational wavenumbers of the functional groups of the Leucaena leucocephala stump biochar produced by activation and modification at 500°C (W5-f) and at 900°C (W912), should be merged in single figure so that difference will, see?
  2. Figure 8. The IR vibrational wavenumbers of the functional groups of the Syzygium samarangense stump biochar produced by activation and modification at 500°C (S5-f) and at 900°C (S912), should be merged in single figure so that difference will, see?
  3. Figure 9. The IR vibrational wavenumbers of the functional groups of the Ziziphus jujuba stump 409 biochar produced by activation and modification at 500°C (I5-f) and at 900°C (I912), should be merged in single figure so that difference will, see?

A: Thank you very much for your valuable suggestions.

The graphs with a temperature of 500 and 900 degrees will be very crowded together, but it is not clear to see. We make a table for a comprehensive explanation (Table 7).

Please see page 15, line 479 to 481

  1. Citation of result discussion will be improved through recent studies in field?

A: Thank you very much for your valuable suggestions.

The development of marco pore biochar can load more probiotics, which is of great help for soil fertilizer conservation, slow release of fertilizers and plant growth.

  1. Conclusion of manuscript will be concise and with quantitative results?

A: Thank you very much for your valuable suggestions.

The conclusion of manuscript already revised. Please see page 16, line 482 to 497

Reviewer 2 Report

This paper is well structured. It has an average significant and novelty. The content is described and contextualized with respect the background. The topic covered is absolutely interesting. The aim of the work is very interesting, the introduction is complete and updated on the topic. The artcle is adeguately referenced.  The results are very clear. The conclusions are consistent with the evidence and  arguments presented addressing the main question that are posed.

Author Response

Manuscript ID: sustainability-1935282: Development of meso- and macro-pore carbonization technology in treating the stumps of representative trees in Taiwan

Many thanks to the reviewer for valuable suggestions. The relevant additions and revisions are as follows:

Comments and Suggestions for Authors

This paper is well structured. It has an average significant and novelty. The content is described and contextualized with respect the background. The topic covered is absolutely interesting. The aim of the work is very interesting, the introduction is complete and updated on the topic. The artcle is adeguately referenced.  The results are very clear. The conclusions are consistent with the evidence and arguments presented addressing the main question that are posed.

A: The loading and encapsulation of probiotics in large-pore biochar is a continuation of this paper, and it will also become an important product of net zero carbon emissions and green materials. Thank you again.

Reviewer 3 Report

1- Please add local name with the technical name of the trees species. 

2- Please add concluding statement of your study in the abstract section. 

3- Authors have mentioned several time about biochar. its better to mention in the article title. 

4- Figure -1. is your own figure or authors used another scientists information. if get from other source then please mention reference. 

5- Authors needs to add some useful literature in your methodology section and introduction section. e.g., Environmental science and pollution research 25 (12), 11875-11883; used in biochar activation methods; Journal of Soils and Sediments 18 (3), 874-886; in biochar preparation method. 

6- Please add Chemosphere 211, 632-639 this reference in the FTIR method as a reference. 

7- Please improve the figure quality of figure.3. 

8- If possible please improve the figure quality of FTIR and add functional groups on relevant peaks. 

9- Please summarize your conclusion section. 

6-  

Author Response

Manuscript ID: sustainability-1935282: Development of meso- and macro-pore carbonization technology in treating the stumps of representative trees in Taiwan

Many thanks to the reviewer for valuable suggestions. The relevant additions and revisions are as follows:

  • Please add local name with the technical name of the trees species. 

A: Thank you very much for your valuable suggestions.

Stumps of three representative trees in Pingtung County, Taiwan: Local name: White Popinac (Leucaena leucocephala), Local name: Wax apple (Syzygium samarangense), Local name: Dates (Ziziphus jujube)

Page 3, line 124 to 125.

  • Please add concluding statement of your study in the abstract section. 

A: Thank you very much for your valuable suggestions, which have been revised. Please See page 1, line 13 to 30.

  • Authors have mentioned several time about biochar. its better to mention in the article title. 

A: Thank you very much for your valuable suggestions.  New article title “Development of meso- and macro-pore carbonization technol-ogy from biochar in treating the stumps of representative trees in Taiwan.”

  • Figure -1. is your own figure or authors used another scientists information. if get from other source then please mention reference. 

A: This photo was taken by ourselves, and the text description refers to Xiaomin Zhu et al. P. 2, line 76 [19].

  • Authors needs to add some useful literature in your methodology section and introduction section. e.g., Environmental science and pollution research 25 (12), 11875-11883; used in biochar activation methods; Journal of Soils and Sediments 18 (3), 874-886; in biochar preparation method. 

A: Thank you very much for your valuable suggestions.

Page 4, line 154 and Page 17, line 570 to 572.

[34]Bashir, S.; Hussain, Q.; Akmal, M.; Riaz, M.; Hu, H.; Ijaz, S. S.; Iqbal, M.; Abro, S.; Mehmood, S.; Ahmad, M. Sugarcane ba-gasse-derived biochar reduces the cadmium and chromium bioavailability to mash bean and enhances the microbial activity in contaminated soil. J. Soils Sediments. 2018, 18, 874-886.

  • Please add Chemosphere 211, 632-639 this reference in the FTIR method as a reference. 

A: Thank you very much for your valuable suggestions.

Page 7, line 573 to 575

[35] Bashir, S.; Hussain, Q.; Shaaban, M.; Hu, H. Efficiency and surface characterization of different plant de-rived biochar for cadmium (Cd) mobility, bioaccessibility and bioavailability to Chinese cabbage in highly contaminated soil. Chemosphere. 2018, 211, 632-639.

  • Please improve the figure quality of figure.3. 

A: Thank you very much for your suggestions, which have been revised. Please See page 9, line 337 to 339.

Figure 3. Effects of low temperature carbonization, high temperature carbonization and activation of oyster shell powder on the specific surface area of ​​three fruit tree biochar samples

  • If possible please improve the figure quality of FTIR and add functional groups on relevant peaks. 

A: Thank you very much for your valuable suggestions. I already improve all FTIR figure to make a table for a comprehensive explanation (Table 7). Please see page 15, line 479 to 481.

9- Please summarize your conclusion section. 

A: Thank you very much for your valuable suggestions.

The conclusion of manuscript already revised. Please see page 16, line 482 to 497

Round 2

Reviewer 1 Report

Manuscript ID: sustainability-1935282: Development of meso- and macro-pore carbonization technology in treating the stumps of representative trees in Taiwan

Still authors are not complete all comments as per given below:

1.      abstract should be informative, quantitative and also remove the grammatical errors?

2.      Novelty of study is missing in the introduction section?

3.      In introduction citation should be improve with recent literatures?

4.      Characterization part of manuscript is very weak so improved through addition such as SEM, EDS, TGA and XRD etc?

5.      How to know Carbonization temperature (oC) of sample shown in table 1 and discuss about that one paragraph?

6.      Citation should be update with recent study? 3.2 The effect of different activation methods on the pore characteristics and specific surface area of biochar samples, compared with KOH and H3PO4, the effective active ingredient in the shell can produce less gas at high temperature and the oyster shell powder is less likely to enter the small pores of the unactivated pre-biochar because of its larger size.

7.      In section 3.2 The effect of different activation methods on the pore characteristics and specific surface area of biochar samples, Author showed these results (Table 3 show that the specific surface area of mesoporous and 267 macropores of Leucaena leucocephala carbon is 62.54 m2 /g; that of Ziziphus jujuba carbon is 268 72.63 m2 /g.) but they should compare with literatures also for scientifically?

8.      in section 3.3 Analysis of the properties of biochar fertilizers after modification, (“After weighing the three kinds of biochar materials after physical activation, 20 times the weight of water was added to mix them evenly, and then 1.0 M acetic acid aqueous solution was gradually added until the pH value of the aqueous solution reached neutrality (pH=7.05).”.) this parameter and also compare with recent literatures?

9.      Figure 7. The IR vibrational wavenumbers of the functional groups of the Leucaena leucocephala stump biochar produced by activation and modification at 500°C (W5-f) and at 900°C (W912), should be merged in single figure so that difference will, see?

10.  Figure 8. The IR vibrational wavenumbers of the functional groups of the Syzygium samarangense stump biochar produced by activation and modification at 500°C (S5-f) and at 900°C (S912), should be merged in single figure so that difference will, see?

11.  Figure 9. The IR vibrational wavenumbers of the functional groups of the Ziziphus jujuba stump 409 biochar produced by activation and modification at 500°C (I5-f) and at 900°C (I912), should be merged in single figure so that difference will, see?

12.  Citation of result discussion will be improved through recent studies in field?

13.  Conclusion of manuscript will be concise and with quantitative results?

Author Response

Manuscript ID: sustainability-1935282: Development of meso- and macro-pore carbonization technology in treating the stumps of representative trees in Taiwan

Thanks so much for your suggestions of the reviewer, and have responded to all the comments, as shown below:

  1. abstract should be informative, quantitative and also remove the grammatical errors?
  • A: Thank you very much for your valuable suggestions, which have been revised. Please See page 1, line 13 to 30.
  1. Novelty of study is missing in the introduction section?
  • A: Thank you very much for the valuable suggestions of the committee members, and the description of novelty has been added with page 2 line53, page 3 line109 and added to recent literatures [30][36][37].

[30] Zhongxin Tan, Carol S.K.Lin, Xiaoyan Ji, Thomas J. Rainey. (2017). Returning biochar to fields: A review. Applied Soil Ecology, 116, 1-11.

[36] Laghari, M.; Naidu, R.; Xiao, B.; Hu, Z.; Mirjat, M. S.; Hu, M.; Kandhro. M. N.; Chen. Z.; Gau. D.; Jogi.; Abudi. Z. N.; Fazal. S. (2016). Recent developments in biochar as an effective tool for agricultural soil management: a review. J. Sci. Food Agric. 2016, 96, 4840-4849.

[37] Semida, W. M.; Beheiry, H. R.; Sétamou, M.; Simpson, C. R.; Abd El-Mageed, T. A.; Rady, M. M.; Nelson, S. D. Biochar implications for sustainable agriculture and environment: A review. S. Afr. J. Bot. 2019, 127, 333-347.

  1. In introduction citation should be improve with recent literatures?
  • A: Thank you very much for the valuable suggestions of the committee members, and the description of novelty has been added. The development of macro porous biochar as a carrier for probiotics is quite an innovative idea. See page 3, line 110 to 121. Added to recent literatures [30][36][37].

[30] Zhongxin Tan, Carol S.K.Lin, Xiaoyan Ji, Thomas J. Rainey. (2017). Returning biochar to fields: A review. Applied Soil Ecology, 116, 1-11.

[36] Laghari, M.; Naidu, R.; Xiao, B.; Hu, Z.; Mirjat, M. S.; Hu, M.; Kandhro. M. N.; Chen. Z.; Gau. D.; Jogi.; Abudi. Z. N.; Fazal. S. (2016). Recent developments in biochar as an effective tool for agricultural soil management: a review. J. Sci. Food Agric. 2016, 96, 4840-4849.

[37] Semida, W. M.; Beheiry, H. R.; Sétamou, M.; Simpson, C. R.; Abd El-Mageed, T. A.; Rady, M. M.; Nelson, S. D. Biochar implications for sustainable agriculture and environment: A review. S. Afr. J. Bot. 2019, 127, 333-347.

  1. Characterization part of manuscript is very weak so improved through addition such as SEM, EDS, TGA and XRD etc?
  • A: Thank you very much for your valuable suggestions.

This paper is "Development of meso- and macro-pore carbonization technology in treating the stumps of representative trees in Taiwan". Therefore, at the stage of this paper, the carbonization process and the properties of pores are analyzed to understand the distribution and proportion of macro-, meso- and micro pores after carbonization and activation of these three representative tree branches in Taiwan. Development of macro porous biochar loaded probiotics as the next stage. Therefore, SEM, EDS, TGA and XRD measurements are not required at this stage.

  1. How to know Carbonization temperature (oC) of sample shown in table 1 and discuss about that one paragraph?
  • A: Thank you very much for your valuable suggestions.

P.5 to 6, line 180 to 224. “3.1 Analysis and discussion of pore characteristics and specific surface area at different carbonization temperatures.” The entire article describes the carbonization temperature and porosity characteristics of the three materials.

  1. Citation should be update with recent study? 3.2 The effect of different activation methods on the pore characteristics and specific surface area of biochar samples, compared with KOH and H3PO4, the effective active ingredient in the shell can produce less gas at high temperature and the oyster shell powder is less likely to enter the small pores of the unactivated pre-biochar because of its larger size.
  • A: Thank you very much for your valuable suggestions.

Two new papers have been added for reference.

[31]Abdul Hafeez, Taowen Pan, Jihui Tian, & Kunzheng Cai. (2022). Modified Biochars and Their Effects on Soil Quality: A Review. Environments 9(5):60, DOI:10.3390/environments9050060.

[32]Wang, Y.; Zhong, B.; Shafi, M.; Ma, J.; Guo, J.; Wu, J.; Ye, Z.; Liu, D.; Jin, H. Effects of biochar on growth, and heavy metals accumulation of moso bamboo (Phyllostachy pubescens), soil physical properties, and heavy metals solubility in soil. Chemo-sphere 219, 510–516.

  1. In section 3.2 The effect of different activation methods on the pore characteristics and specific surface area of biochar samples, Author showed these results (Table 3 show that the specific surface area of mesoporous and 267 macropores of Leucaena leucocephala carbon is 62.54 m2 /g; that of Ziziphus jujuba carbon is 268 72.63 m2 /g.) but they should compare with literatures also for scientifically?
  • A: Thank you very much for your valuable suggestions.

Two new papers have been added for reference.

[31]Abdul Hafeez, Taowen Pan, Jihui Tian, & Kunzheng Cai. (2022). Modified Biochars and Their Effects on Soil Quality: A Review. Environments 9(5):60, DOI:10.3390/environments9050060.

[32]Wang, Y.; Zhong, B.; Shafi, M.; Ma, J.; Guo, J.; Wu, J.; Ye, Z.; Liu, D.; Jin, H. Effects of biochar on growth, and heavy metals accumulation of moso bamboo (Phyllostachy pubescens), soil physical properties, and heavy metals solubility in soil. Chemo-sphere 219, 510–516.

  1. in section 3.3 Analysis of the properties of biochar fertilizers after modification, (“After weighing the three kinds of biochar materials after physical activation, 20 times the weight of water was added to mix them evenly, and then 1.0 M acetic acid aqueous solution was gradually added until the pH value of the aqueous solution reached neutrality (pH=7.05).”.) this parameter and also compare with recent literatures?
  • A: Thank you very much for your valuable suggestions.

Biochar for soil improvement or combination of probiotics, biochar must have a suitable pH. For biochar-coated probiotics, the pH should be neutral, so the pH adjustment of relevant biochar can refer to this book [33].

[33]Balwant Singh, Dolk MM, Qinhua Shen, Marta Camps Arbestain. (2017). Chapter 3. Biochar pH, electrical conductivity and liming potential. In book: Biochar: A Guide to Analytical Methods (pp.23-38).

  1. Figure 7. The IR vibrational wavenumbers of the functional groups of the Leucaena leucocephala stump biochar produced by activation and modification at 500°C (W5-f) and at 900°C (W912), should be merged in single figure so that difference will, see?
  • A: Thank you very much for your valuable suggestions.

A1. The graphs with a temperature of 500 and 900 degrees will be very crowded together, but it is not clear to see. We make a table for a comprehensive explanation (Table 7). Please see page 15, line 47 to 479

A2. Figures 7, 8, and 9 are compiled into one big picture, which is easier to compare and contrast (Figure 7). Please see page 13, line 422 to 425

  1. Figure 8. The IR vibrational wavenumbers of the functional groups of the Syzygium samarangense stump biochar produced by activation and modification at 500°C (S5-f) and at 900°C (S912), should be merged in single figure so that difference will, see?
  • A: Thank you very much for your valuable suggestions.

A1. The graphs with a temperature of 500 and 900 degrees will be very crowded together, but it is not clear to see. We make a table for a comprehensive explanation (Table 7). Please see page 15, line 477 to 479

A2. Figures 7, 8, and 9 are compiled into one big picture, which is easier to compare and contrast (Figure 7). Please see page 13, line 422 to 425

  1. Figure 9. The IR vibrational wavenumbers of the functional groups of the Ziziphus jujuba stump 409 biochar produced by activation and modification at 500°C (I5-f) and at 900°C (I912), should be merged in single figure so that difference will, see?
  • A: Thank you very much for your valuable suggestions.

A1. The graphs with a temperature of 500 and 900 degrees will be very crowded together, but it is not clear to see. We make a table for a comprehensive explanation (Table 7). Please see page 15, line 477 to 479

A2. Figures 7, 8, and 9 are compiled into one big picture, which is easier to compare and contrast (Figure 7). Please see page 13, line 422 to 425

  1. Citation of result discussion will be improved through recent studies in field?
  • A: Thank you very much for your valuable suggestions.

The development of macro- pore biochar can load more probiotics, which is of great help for soil fertilizer conservation, slow release of fertilizers and plant growth.

  1. Conclusion of manuscript will be concise and with quantitative results?
  • A: Thank you very much for your valuable suggestions.

The conclusion of manuscript already revised. Please see page 15, line 480 to 495

Round 3

Reviewer 1 Report

Now authors respond all comments given by the reviewer and it will accepted for publication,